# Contribution of LAT1-4F2hc in Urological Cancers via Toll-like Receptor and Other Vital Pathways

**DOI:** 10.3390/cancers14010229

**Published:** 2022-01-04

**Authors:** Xue Zhao, Shinichi Sakamoto, Maihulan Maimaiti, Naohiko Anzai, Tomohiko Ichikawa

**Affiliations:** 1Department of Urology, Chiba University Graduate School of Medicine, Chiba 260-8670, Japan; abesusuki@126.com (X.Z.); ichikawa@vmail.plala.or.jp (T.I.); 2Department of Urology, Tongren Hospital, Shanghai Jiao Tong University School of Medicine, Shanghai 200336, China; 3Department of Tumor Pathology, Chiba University Graduate School of Medicine, Chiba 260-8670, Japan; marghulanmaimaiti@gmail.com; 4Department of Pharmacology, Chiba University Graduate School of Medicine, Chiba 260-8670, Japan; anzai@chiba-u.jp

**Keywords:** L-type amino acid transporter 1 (LAT1, SLC7A), 4F2 cell-surface antigen heavy chain (4F2hc, SLC3A2), urinary system tumors, diagnosis, targeted therapy

## Abstract

**Simple Summary:**

LAT1-4F2hc complex is an important amino acid transporter. It mainly transports specific amino acids through the cell membrane, provides nutrition for cells, and participates in a variety of metabolic pathways. LAT1 plays a role in transporting essential amino acids including leucine, which regulates the mTOR signaling pathway. However, the importance of SLCs is still not well known in the field of urological cancer. Therefore, the purpose of this review is to report the role of the LAT1-4F2hc complex in urological cancers, as well as their clinical significance and application. Moreover, the inhibitor of LAT1-4F2hc complex is a promising direction as a targeted therapy to improve the treatment and prognosis of urological cancers.

**Abstract:**

Tumor cells are known for their ability to proliferate. Nutrients are essential for rapidly growing tumor cells. In particular, essential amino acids are essential for tumor cell growth. Tumor cell growth nutrition requires the regulation of membrane transport proteins. Nutritional processes require amino acid uptake across the cell membrane. Leucine, one of the essential amino acids, has recently been found to be closely associated with cancer, which activate mTOR signaling pathway. The transport of leucine into cells requires an L-type amino acid transporter protein 1, LAT1 (SLC7A5), which requires the 4F2 cell surface antigen heavy chain (4F2hc, SLC3A2) to form a heterodimeric amino acid transporter protein complex. Recent evidence identified 4F2hc as a specific downstream target of the androgen receptor splice variant 7 (AR-V7). We stressed the importance of the LAT1-4F2hc complex as a diagnostic and therapeutic target in urological cancers in this review, which covered the recent achievements in research on the involvement of the LAT1-4F2hc complex in urinary system tumors. In addition, JPH203, which is a selective LAT1 inhibitor, has shown excellent inhibitory effects on the proliferation in a variety of tumor cells. The current phase I clinical trials of JPH203 in patients with biliary tract cancer have also achieved good results, which is the future research direction for LAT1 targeted therapy drugs.

## 1. Introduction

Continuous proliferative signaling is the main feature of malignant tumors [1]. These signals trigger tumor cells to divide, causing tumor cells to grow rapidly in an uncontrollable way. Among all of these nutrients, Eagle discovered in 1955 that essential amino acids (EAA) were required for cell growth in vitro [2]. Later, studies found that the uptake of EAA in malignant tumor cells was higher than in normal tissues [3,4,5]. After being delivered into the cells, these amino acids were utilized to make proteins, nucleic acids, lipids, and ATP. Cancer cells have higher up-regulated transporters that facilitate the entrance of exogenous amino acids into cells, compared to normal cells, and the steady acquisition of amino acids by cancer cells is important for cancer growth [6]. HATs (heteromeric amino acid transporters) are a special type of solute transporter. They are made up of two subunits, one heavy and one light, that are linked by a conserved disulfide bond [7]. The heavy subunit is a member of the SLC3 family, whereas the light subunit belongs to the SLC7 family. 

The SLC3 family now includes two glycoproteins (rBAT (SLC3A1)) and 4F2hc (SLC3A2, also known as CD98) [7]. Heavy subunits of the SLC3 family, such as 4F2hc, were discovered in 1998 and are necessary for the proper trafficking of the heterodimer to the plasma membrane [8]. 

Regarding the SLC7 family, Kanai first isolated a cDNA from rat C6 glioma cells through expression cloning in 1998. The cDNA encodes a new Na +-independent neutral amino acid transporter called LAT1 [9]. In 1999, Kanai’s team further isolated a cDNA from the rat small intestine, which encodes another transporter called LAT2 [10]. The former two proteins belong to the solute carrier family 7 (SLC7). After that, LAT3 [11] and LAT4 [12] were gradually discovered. These two belong to the SLC43 family. The L-type amino acid transporter, which consists of all former four subunits (LAT1-4), is an important pathway for EAA to enter the cell. Subsequently, Wang found that (18) F-labeled fluoroalkyl phenylalanine derivatives as PET tracers were more likely to bind to LAT1 in tumors, and the specific accumulation of this tracer in tumor cells suggested that LAT1 was expressed in a large number of malignant tumors, thus preliminarily revealing the close relationship between LAT1 and malignant tumors [13]. In 2016, U.S. Food and Drug Administration approved trans-1-amino-3-18F-fluorocyclobutanecarboxylic-acid (anti-[18F]-FACBC) PET for the detection of prostate cancer in patients with elevated prostate-specific-antigen following curative treatment [14]. LAT1 is known to be the primary target of FACBC [15]. The usefulness of LAT1 in PET imaging has already been validated in clinical practice.

In a previous extensive review, Wang reported that among the four LAT transporters, LAT1 (SLC7A5) is overexpressed in various cancers, which is more widespread than the other three LAT transporters [3]. Subsequent research intensified and found that the complex composed of 4F2hc and LAT1 played play a key role in the occurrence and development of multiple human tumors. How to block the transport of nutrients by HATs to malignant tumor cells to achieve the purpose of inhibiting the occurrence and development of malignant tumor cells is an attractive research topic.

However, the importance of SLCs is still not well known in the field of urological cancer. In particular, LAT1 is a target of FACBC PET [15], which has important imaging implications in prostate cancer, following PSMA PET. Recently, 4F2hc, which binds to LAT1, has been identified as a specific downstream signal of AR-V7, a cause of castration resistance [16]. JPH203, a specific inhibitor of LAT1, has already completed Phase I clinical trials in Japan and may be applied to prostate cancer in the future [17].

Therefore, in this review, we summarized the latest advances in research on the role of the LAT1-4F2hc complex in urinary system tumors and emphasized the importance of the LAT1-4F2hc complex as a diagnostic and therapeutic target in urinary system tumors.

## 2. LAT1-4F2hc Complex and Structural Characteristics

LAT1 is made up of two layers of 12 putative transmembrane segments (TMs). TM1, TM3, TM6, TM8, and TM10 make up the inner layer, which is encircled by the outer layer. The outer layer is made up of TM2, TM4, TM5, TM7, TM9, TM11, and TM12. LAT1’s N- and C-terminal ends are intracellularly localized, whereas 4F2hc’s N- and C-terminal ends are intracellularly and extracellularly localized. The contact between 4F2hc and LAT1 is limited to one side of LAT1, while TM1 and TM6 of LAT1 are construction switches, which are essential for the alternate entry transport mechanism of the LeuT-fold transporters, and their positions are far away from the coordination of 4F2hc. Therefore, 4F2hc seems to stabilize the scaffold domain of LAT1 in the membrane, which may contribute to the local conformational shift of gating elements (such as TM1, TM2, TM6, and TM10) during alternate entry cycles [18,19,20] (Figure 1A–C).

According to the structure of LAT1-4F2hc heterodimeric amino acid transporter protein complex, 4F2hc had only one transmembrane helix that seemed to be unable to form a transmembrane transporter pore. It shows that 4F2hc has a lack of amino acid transport activity. In contrast, LAT1 shows a typical membrane transport protein helical bundle structure. That is the reason why past studies have reported that LAT1 is the only sole transport-competent unit, and 4F2hc does not play any significant role in the internal transport function [22]. Now, there are different views about it. Glycoprotein 4F2hc acts as a molecular chaperone to make LAT1 the final location on the cell membrane [23]. In the absence of 4F2hc, LAT1 is present in the intracellular compartment, while 4F2hc can independently reach the plasma membrane [8,23]. In the presence of LAT1, the surface expression pattern of 4F2hc changes, restricting it to cell-cell adhesion sites [23]. 4F2hc is necessary for the transport of LAT1 to the plasma membrane, and LAT1 is believed to determine the transport properties of heterodimers. It is obvious that LAT1 and 4F2hc cannot work alone without each other. Meanwhile, in many forms of cancer, increased 4F2hc expression levels have been linked to a worse prognosis in several studies [24,25,26,27]. The most critical structure involved in the interaction of the complex is the disulfide bond between the two proteins [8,23]. The functional role of the disulfide bond is still unclear. It does not seem to be involved in the ectopic of the two proteins to the membrane, nor in the transport of amino acids. However, recent studies have shown that disulfide bonds are important for regulating 4F2hc-related cation channels [28].

LAT1-4F2hc heterodimeric amino acid transporter protein complex is a transmembrane transporter that independent of Na+ and pH. It imports large neutral amino acids (such as leucine and phenylalanine) for intracellular amino acid exchange (e.g., glutamine) [7,29], which are abundant in cells that require a constant supply of amino acids, such as nerve cells, activated T cells, placental cells, glial cells, and blood-brain barrier (BBB) endothelial cells [9,30,31]. In BBB, LAT1-4F2hc complex is stereospecific (L > D) [32]. Compared with LAT1 in peripheral tissues [33], it has a higher affinity for amino acids. Studies have shown that the affinity of LAT1 to intracellular amino acids is higher than that of extracellular amino acids, demonstrating that the quantity of intracellular substrate regulates LAT1 transport rate [34]. Due to its own transport characteristics, the LAT1-4F2hc complex often plays a key role in drug absorption, distribution and toxicity by mediating drug transmembrane transport, and often represents unexpected off-target of drugs [35].

## 3. LAT1/4F2hc and Human Diseases (Pain & Inflammation) 

Existing studies have found that LAT1-4F2hc complex is widely associated with human diseases, such as inflammation, pain, hypoxia, and tumors [36,37,38]. 

Inhibition of LAT1 eliminated mTORC1 activation, plasmablast differentiation, and CpG (toll-like receptor TLR9 ligand)-stimulated B cell production of IgG and inflammatory cytokines. The influx of L-leucine through LAT1 regulates the activity of mTORC1 and the immune response of human B cells [37,38]. Among the most common nociceptive pathways, LAT1 may be a feasible new target for pain. LAT1 expression and regulation link it to key cell types and pathways related to pain. LAT1 regulates the Wnt/frizzled/β-catenin signal transduction pathway. The LAT1-4F2hc complex may also be involved in pain pathways related to T cells and B cells. The expression of LAT1 induces the activation of the mammalian target of rapamycin (mTOR) signal axis, which is related to inflammation and neuropathic pain. Similarly, hypoxia and tumors can induce the activation of hypoxia-inducible factor 2α, which not only promotes the expression of LAT1 but also promotes the activation of mTORC1 [36]. As the common node of the T cell, B cell, and mTOR pathway, LAT1-4F2hc plays a vital role in human diseases. It has also received increasing attention as an important target for autoimmune diseases, chronic pain diseases, and tumors.

## 4. LAT1/4F2hc and Tumors

Many tumor cells lines [39,40,41] and human malignancies, such as breast, prostate, lung, colorectal, and gliomas [42,43,44,45,46,47], have high levels of LAT1 expression. In these tumors, LAT1 plays an important role in growth and survival. RNA interference (RNAi) [44,48,49,50,51] and genetic disruption by zinc fingers nucleases-mediated [52] LAT1-knockout in cancer cells caused that leucine absorption and cell proliferation were both inhibited. As a result, LAT1 is being evaluated as a potential therapeutic target for reducing cancer cell growth and proliferation [53,54].

Similarly, in human neoplasms such as prostate cancer, gastric cancer, lung pleomorphic carcinoma, and neuroendocrine carcinoma, 4F2hc expression is upregulated [24,27,41,55]. Increased 4F2hc expression is linked to a worse chance of survival, cell proliferation, and metastasis [56]. Since 4F2hc binds with LAT1 on the membranous surface of cancer cells, these results are not difficult to understand.

The LAT1-4F2hc complex is also closely related to tumor glutamine metabolism. The amount of glutamine required by cancer cells exceeds the supply produced by endogenous synthesis, resulting in the up-regulation of glutamine metabolism in many carcinogenic changes. LAT1-4F2hc complex controls the flux of glutamine and other amino acids involved in glutaminolysis and glutamine-regulated homeostasis [35]. LAT1-4F2hc complex exchanges Gln for leucine and other amino acids, which can lead to mTOR activation.

By influencing the mammalian target protein of rapamycin complex 1 (mTORC1), the amino acid leucine has been demonstrated to increase protein synthesis and accelerate cell development, whereas LAT1 has been linked to mTORC1 signaling and, as a result, cancer progression [6,57]. 

In cancer cells, however, LAT1 not only boosts mTORC1 activity but also enhances MYC and EZH2 signaling. Through the AKT, MAPK, and cell-cycle related P21 and P27 signal pathways, 4F2hc has been demonstrated to affect cancer cell proliferation. The expression of 4F2hc and LAT1 is reportedly codependent, and the downregulation of either subunit destabilizes the partner [8]. (Figure 2, Table 1).

## 5. LAT1/4F2hc and Urological Tumors

### 5.1. LAT1/4F2hc and Prostate Cancer

LAT1-4F2hc complex plays an important role in growth and survival in PCa cells. Sakata used LAT1 as a biomarker for highly malignant prostate cancer in 2009 [47]. The increased expression of LAT1 in prostate cancer is a new independent biomarker of high malignancy that can be used to estimate prognosis in conjunction with the Gleason score [47].

Trans-1-amino-3-18F-fluorocyclobutanecarboxylic-acid (anti-[18F]-FACBC) is an amino acid PET tracer, which shows good prospects in visualizing PCa [91]. The tracer is used for the evaluation of l-amino acid transport, LAT1 is known to be the primary target of FACBC [15]. In 2016, 18F-FACBC has been approved by the US Food and Drug Administration (FDA) and the European Commission (EC) to detect PCa in patients with elevated PSA after previous treatments [14]. Approval is based on encouraging diagnostic performance and histologically confirmed data from patients with biochemical relapse [92]. Recently, it was included in the National Comprehensive Cancer National (NCCN) guidelines for the treatment of patients with recurrent PCa. The usefulness of LAT1 in PET imaging has already been validated in clinical practice.

Wang reported [57] that when LAT activity was inhibited, activating transcription factor 4–mediated overexpression of amino acid transporters such as ASCT1, ASCT2, and 4F2hc occurred, all of which were regulated by the androgen receptor. LAT suppression inhibited M-phase cell cycle genes regulated by E2F family transcription factors, including UBE2C, CDC20, and CDK1, which are important castration-resistant prostate cancer regulators. In silico analysis of BCH-downregulated genes revealed that in metastatic castration-resistant prostate cancer, 90.9 percent are statistically significantly upregulated. Finally, in vivo, LAT1 knockdown decreased tumor development, cell cycle progression, and spontaneous metastasis in xenografts [57].

Patel studied the functional characterization and molecular expression of large neutral amino acids of LAT1 in prostate cancer PC-3 cells [93]. It proves that LAT1 is mainly responsible for the uptake of large neutral amino acids and has functional activity in PC-3 cells. The fact that Ile-quinidine generates a considerable increase in absorption compared to quinidine suggests that LAT1 could be used to improve the cellular permeability of poorly cell-permeable anticancer medicines. This cell line can also be utilized as an in vitro model to investigate the interaction of large-scale neutral amino acid conjugated pharmaceuticals with the LAT1 transporter [94].

In PCa cell lines, DU145 cells had the highest levels of 4F2hc protein expression, followed by PC-3 and C4-2 cells. In C4-2 and DU145 cells, 4F2hc expression was found to be substantially greater than LAT1 expression. Cell growth, migration, and invasion are all inhibited by Si4F2hc. 4F2hc and LAT1 expression in PCa tissue and association with clinical variables. The expression levels (4F2hc and LAT1/high and low) are associated with various tumor prognoses [24]. The data from the same study [24] revealed that SKP-2 is a downstream and particular target gene of 4F2hc. SKP-2 is associated with cell cycle, DNA replication, and cell division.

#### 5.1.1. AR and LAT1-4F2hc Complex in CRPC (AR/AR-V7 and 4F2hc Promotes the Development of CRPC)

Xu reported that the up-regulation of LAT1 during anti-androgen therapy promotes the progression of PCa cells [44]. In hormone-resistant prostate cancer cell lines, LAT1 was shown to be substantially expressed. Knocking down LAT1 in LNCaP and C4-2 cells can drastically reduce cell proliferation, migration, and invasion. In patients receiving androgen deprivation therapy, high LAT1 expression was linked to a significantly shorter prostate-specific antigen recurrence-free survival [44].

Another study demonstrated a potential relationship between AR-V7 and 4F2hc [16]. AR-V7 activates downstream target genes in the absence of androgens. 4F2hc (SLC3A2) is one of the downstream target genes of AR-V7. AR-V7 gene knockdown leads to a decrease in the level of H3K27ac at the 4F2hc locus. The decrease in the expression of 4F2hc indicates that AR-V7 has a certain effect on the activation of 4F2hc expression. In clinical samples, the expression level of 4F2hc in benign lesions and primary PCa tissues was low, while the expression level of 4F2hc in CRPC tissues was significantly increased. The expression of 4F2hc in PCa patients with high AR-V7 expression is higher than that in PCa patients with low AR-V7 expression.

When LNCaP and LNCaP95 cell lines were treated with siRNA against 4F2hc, cellular growth was significantly suppressed [16]. Down-regulation of 4F2hc inhibited cell proliferation through apoptosis and cell senescence [16].

#### 5.1.2. LAT1/4F2hc Expression Is Coordinately Regulated during Prostate Cancer Progression (HSPC to CRPC)

Not all prostate tumor cell lines are closely related to LAT1. Otsuki found that LNCaP cells mainly express LAT3, and LAT1 was primarily expressed in DU145 and PC-3 cells [95]. Xu’s research also gave similar results [44]. LAT3 was abundantly expressed in AR-expressing LNCaP and C4-2 cells, whereas it was barely expressed in AR-negative PC3 and DU145 cells, according to Rii’s study [96]. 

Wang [97] reported the fact that LAT1 is highly expressed in androgen-insensitive PC-3 cells but LAT3 is highly expressed in androgen-sensitive LNCaP cells could be explained by transcriptional regulation of LAT1 and LAT3 expression. Changes in the microenvironment, such as starvation or hormone deprivation, can promote cancer formation and alter LAT1 and LAT3 expression. Reduced androgen receptor signaling may result in decreased LAT3 expression and, as another result, higher LAT1 expression. The results were confirmed by both nude mice samples and human samples [97]. LAT3 expression was higher in amplified AR patients. In a dose-dependent way, DHT stimulation enhanced LAT3 expression. Bicalutamide inhibited the effect of DHT on LAT3 expression. DHT treatment significantly boosted AR expression, which was reduced by bicalutamide [96] (Figure 3).

Since high AR-V7 expression is one of the most common features of CRPC, AR-V7 expression following LH-RH therapy up-regulates the 4F2hc expression [16].

Based on the above evidence, LAT1/4F2hc can be independent PCa biomarkers and therapeutic targets, respectively. They can also collectively influence the transformation of PCa to CRPC and promote both progressions through the mentioned pathways below (Figure 4).

### 5.2. LAT1/4F2hc and Renal Cancer

There are few studies on LAT1 and renal clear cell carcinoma. In 2013, Hironori [42] studied the expression of LAT1, LAT2, LAT3, LAT4, and 4F2hc mRNA in clear cell renal cell carcinoma tissues. It was found that the expression of LAT1 mRNA in tumor tissue was considerably higher than in non-tumor tissue, but the expression of LAT2 and LAT3 mRNA was lower. There was no difference in LAT4 and 4F2hc mRNA expression between tumor and non-tumor tissues. Poorly differentiated tumors, local invasion, microvascular invasion, and metastasis are all linked to increased LAT1 mRNA expression. Higher LAT1 mRNA levels in the blood are linked to a shorter total survival period. Phosphorylated S6 ribosomal protein levels are related to metastatic potential. The level of phosphorylated S6 ribosomal protein is positively linked with the expression of LAT1 mRNA in primary cancers [42].

Higuchi investigated the LAT1 expression profile in RCC tissues as well as its relationships with clinical variables retrospectively [98]. Most of the tissues (92 percent) had cancer-associated LAT1 expression. Patients with high LAT1 expression levels had lower overall survival and progression-free survival than those with low LAT1 expression levels, and these correlations were confirmed by univariate and multivariate analyses [98].

Tumors grow and evolve through continuous crosstalk with the surrounding microenvironment. New evidence shows that angiogenesis and immunosuppression often occur simultaneously to deal with this crosstalk [99]. At present, one strategy to achieve a higher clinical response in the study of renal cell carcinoma is to produce a more effective anti-tumor contraction by combining multiple immune checkpoints. However, the toxicity profile is higher [100]. T cells can shape tumor blood vessels and tumor endothelial cells, prevent the recruitment and infiltration of effector immune cells while remodeling ECM, and further inhibit the migration and infiltration of functional immune cells. The tumor vascular system actively participates in immunosuppression. The abnormal pathophysiological mechanism of tumor vessels can lead to the production of immunosuppressive molecules and inhibit the function of effective T cytotoxic cells. At the same time, the production of chemokines and cytokines promotes the differentiation and activation of immunosuppressive cells. These cells can also inhibit the activity of cytotoxic T cells. On the contrary, in the blood vessels, these mechanisms also down-regulate a variety of adhesion molecules, which are very important for the rolling, adhesion, and transport of T cells into the cancer environment. The normal tumor vascular system can improve T cell infiltration, enhance immune response, stop the immunosuppressive environment, make it a more immunoactivated phenotype, and work together with cancer immunotherapy. Anti-vascular endothelial growth factor receptor (anti-VEGFR) is the first to realize the normalization and functional recovery of tumor vascular system by tissue perfusion and reducing intratumoral hypoxia [99]. In the current studies of cancers [71,101,102], angiogenesis in vitro/in vivo experiments was inhibited by eliminating the function or expression of LAT1. It regulates proliferation, translation, and angiogenesis VEGF-A signal [102]. LAT1 is a central transporter of essential amino acids in human umbilical vein endothelial cells [103]. LAT1 also mediated miR-126 on primary human lung microvascular endothelial cells’ angiogenesis via regulation of mTOR signaling [104]. LAT1 expression correlated significantly with CD98, VEGF, CD34 expression, and microvessel density in the primary and metastatic sites of tumors [41,81,105,106,107,108]. VEGF and CD34 are also related to angiogenesis. These studies further revealed the dual role of LAT1-4F2hc in tumor cells and stromal endothelial cells. The therapeutic inhibition of LAT1-4F2hc may provide an ideal choice for strengthening anti-angiogenesis therapy. Lat1-4f2hc is a potential therapeutic target for anti-tumor angiogenesis and maintenance of the normal vascular system. Therefore, the combination of antiangiogenic therapy and immunotherapy seems to have the potential to break the balance of the tumor microenvironment and improve the treatment response of renal cell carcinoma. It can be a novel paradigm to envision tailored approaches in renal cell-carcinoma and other urological tumors.

### 5.3. LAT1/4F2hc and Bladder Cancer

In 2002, Kyung reported the characterization of the system L amino acid transporter in T24 cells [93]. T24 human bladder cancer cells express LAT1 and its associated protein 4F2hc in the plasma membrane, however, T24 cells do not express the other system L isoform LAT2. The majority of [14C]L-leucine uptake is mediated by LAT1 in T24 cells [93]. 

Baniasadi [109] reported the gene expression profile of inhibiting LAT1 in T24 human bladder cancer cells. BCH influences the expression of a vast number of genes involved in cell survival and physiological function, according to researchers. These findings contribute to a better understanding of the intracellular signaling pathways involved in cell growth suppression produced by LAT1 inhibitors, which could be utilized as a target for anticancer drug development [44,109]. 

Maimaiti studied the expression profile and functional role of LAT1 in bladder cancer [110]. This is the first study to show that LAT1 plays a role in bladder cancer, and it also found IGFBP-5 to be a new downstream target for inhibiting LAT1. High LAT1 expression was found to be an independent predictive factor for overall survival in multivariate analysis. Patients with high LAT1 and IGFBP-5 expression had a significantly lower overall survival than those with low expression. LAT1 levels are linked to pathological alt staging, LDH, and NLR. In vitro, inhibiting LAT1 prevents cell proliferation, migration, and invasion. In aggressive BC patients, IGFBP-5 expression is also linked to a better prognosis [110] (Table 2).

## 6. Inhibitors of LAT1/4F2hc and Targeted Therapy

Due to its own transport characteristics of the SLC family, the LAT1-4F2hc complex often plays a key role in drug absorption, distribution and toxicity by mediating drug transmembrane transport [35]. However, only a small number of SLCs have been locked by drugs or chemical probes till now. Three main factors hinder the development of new chemical entities that can regulate SLC activity. First, most studies on this super population are relatively insufficient, and the biological functions or substrates of many SLCs are still unclear. Second, there is a lack of high-quality biological tools, specific, and reliable reagents and special databases. Finally, the number of functional analyses required to study such diverse objectives is still limited [113]. It is reported radioligand uptake assays have been widely employed to study LAT1 [114], but the radioligand uptake assays cannot distinguish inhibitors from substrates. The LAT1-4F2hc complex is overexpressed in many cancer cells and is thought to be a viable anticancer therapeutic target since inhibiting it reduces cancer cell viability dramatically.

BCH and JPH203 are LAT1-4F2hc complex inhibitors that have been studied extensively. BCH is a non-metabolic leucine analogue. In 2006, Baniasadi [109] found that BCH has an impact on the expression of many genes involved in cell survival and physiological activity. These data help to understand the intracellular signal transduction of cell growth inhibition induced by LAT1 inhibitors and can be used as a candidate for anticancer drug therapy [109]. Later studies proposed the use of N-butyl-N- (4-hydroxybutyl) nitrosamine (BBN) treatment to induce high expression of LAT1/4F2hc in rat bladder cancer cells [101] and proposed some directions for anti-LAT1/4F2hc drugs. JPH203 was discovered by Oda in 2010 and was originally known as KYT-0353 [115]. JHP203 is a highly selective LAT1 inhibitor produced by synthetic chemistry and in vitro screening based on triiodothyronine (T3). JPH203 showed excellent selective inhibition of LAT1 and showed potential as a novel antitumor agent. JPH203 interferes with constitutive activation of mTORC1 and Akt, reduces c-MyC expression, and triggers a folding protein response mediated by CHOP transcription factors associated with cell death [116]. Since then, several studies have confirmed that JPH203 has an impressive inhibitory effect on the growth of common tumor cells, such as colon cancer [115,117], gastric carcinoma [64], medulloblastoma [118], osteosarcoma [119], thyroid cancer [120,121], endocrine-resistant breast cancer [122], pituitary tumor [123], head and neck cancer cells [124], and T-cell Acute lymphoblastic leukemia (T-ALL)/lymphoma (T-LL) cells [116], etc.

In terms of urinary tumors, Maimaiti [110] found that in bladder cancer cells JPH203 inhibits the absorption of leucine by >90%. JPH203 inhibits the phosphorylation of MAPK/Erk, AKT, p70S6K, and 4EBP-1. JPH203 inhibits IGF-mediated igfb5 expression and AKT phosphorylation [110].

In the area of RCC, Higuchi [98] has tested the effects of JPH203 on RCC-derived Caki-1 and ACHN cells. JPH203 suppressed the proliferation of various cell types in a dose-dependent manner. According to the findings, the migration and invasion operations were stifled by JPH203 [98].

In the area of PCa, Otsuki [95] found that LAT1 was primarily expressed in DU145 and PC-3 cells. BCH or JPH203 inhibited leucine uptake and cell proliferation in a dose-dependent manner [95]. A Phase I clinical study found that JPH203 was well-tolerated and provided promising activity against biliary tract cancer [17]. The authors are currently planning Phase I and II study of JPH203 in CRPC [17].

These studies also show the potential of JPH203 for the treatment of urological cancers. 

In 2021, Yan [125] synthesized three LAT1 inhibitors, JX-075, JX-078, and JX-119, and used cryo-EM to solve the inhibitors’ complex structures with the LAT1-4F2hc complex. They also solved the LAT1-4F2hc complex coupled with Diiodo-Tyr’s cryo-EM structure. LAT1 is found in an outward-occluded conformation in all the combinations of these complexes. These structures might reflect two distinct inhibitory processes, giving significant information for medication development in the future [125].

Of particular interest is the first Phase I clinical trial of JPH203 [17]. Although several studies have demonstrated that JPH203 can inhibit leucine uptake by tumor cells and show concentration-dependent cytotoxicity in vitro or good results in transplanted tumor models, Phase I clinical trial in humans is a milestone. Okano assessed dose-limiting toxicity in the first cycle using the 3 + 3 design. Seventeen Japanese patients with advanced solid tumors were enrolled and treated daily with JPH203 intravenously for 7 days. The maximum safe tolerated dose of JPH203 was defined as 60 mg/m^2^. The suitable RP2D is 25 mg/m^2^. Partial response was observed in one biliary tract cancer (BTC) patient at 12 mg/m^2^, and disease control was achieved in three of the six BTC patients at both the 12 mg/m^2^ and 25 mg/m^2^ levels. The disease control rate of BTC was 60%. The JPH203 molecule is predominantly metabolized into Nac-JPH203 by N-acetyltransferase 2 in liver cells [126]. Patients’ N-acetyltransferase 2 phenotype (rapid/non-rapid) was found to predict the safety and efficacy of JPH203. A lower Nac-JPH203/JPH203 ratio is critical for maximizing the anti-tumor effect of JPH203 [17].

Of course, there are still some deficiencies and limitations in the study of urinary tumors and LAT1-4F2hc complexes mentioned above.

In BBN-induced bladder cancer, LAT1-4F2hc was not expressed by porous endothelial cells. Whether LAT1-4F2hc expression depends on endothelial cell structure is unclear. Fenestration of microvascular endothelial cells is not a stable event, because endothelial cells with fenestration in BBN-induced rat bladder cancer were transformed into endothelial cells without fenestration 5 min after injection of VEGF inhibitor, and fenestration recovered 30 min later [101]. The molecular mechanisms of amino acid transport in normal and tumor microvascular endothelial cells need further study. However, the LAT1-4F2hc complex is closely related to angiogenesis [41,71,81,101,102,105,106,107,108]. This makes it possible for the LAT1-4F2hc complex to improve the effectiveness of cancer immunotherapy by improving immune vascular crosstalk [99].

In prostate cancer-related experiments, although downregulation of LAT1 and LAT3 in tumor cells inhibits the growth of prostate cancer cells, it remains to be determined what other mechanisms of prostate cancer resistance can be triggered by targeting LAT1 (such as activation of ATF4).

Most of the studies were conducted in vitro, not in vivo. Although the phase I clinical trial of JPH203 against biliary tract cancer has achieved good results, the clinical trial has not yet involved any urinary tumors. In addition, the number of patients included in some studies is relatively small, or the follow-up time is not long, and the prognostic impact of LAT1 inhibition on tumor patients with different stages has not been thoroughly solved. Most of the specimens studied are in vitro tumor specimens after surgery, and the expression of LAT1-4F2hc in early tumors and its influence on tumors are also a key link that needs to be studied.

Finally, targeted therapy of LAT1-4F2hc does not directly kill cancer cells, but blocks amino acid transport, resulting in loss of nutritional basis and self-apoptosis of cancer cells. This has led some investigators to suggest that targeting LAT1-4F2hc is more suitable for slow-progressing tumors. Therefore, further studies are needed to obtain more evidence that LAT1-4F2HC therapy is also suitable for highly aggressive and rapidly progressing tumors. 

## 7. Conclusions

### Significant Contribution of the LAT1-4F2hc in Urological Cancers

These studies and experiments above are helping us to understand how cancer cells metabolize differently from normal cells, as well as the therapeutic targets that could be interfered with in these different metabolisms of proliferation. The abnormal proliferation of tumor cells usually depends on the nutrient microenvironment generated by these abnormal metabolic patterns. Recognizing and blocking the nutrient absorption pathways of malignant tumors are usually the key points in the diagnosis and treatment of malignant tumors. The LAT1-4F2hc complex is such a target with both diagnostic and therapeutic significance. 

The LAT1-4F2hc complex mediates a variety of pathways, such as T cells, B cells, and mTOR pathways, and is also closely related to Toll-like receptors and vascular endothelial growth factors. This has caused the LAT1-4F2hc complex to become a common factor in many diseases, such as autoimmune diseases, pain, tumors.

The clinical significance of the LAT1-4F2hc complex in urinary cancer has gradually begun to be explored and confirmed, just like in other tumor cells. LAT1-4F2hc upregulation seems to be a common phenomenon in cancers. It is a reliable tumor biomarker and the target of imaging tracer, which can be used for the diagnosis and prognosis of urinary malignant tumors. It is also a meaningful therapeutic target. In fact, great efforts have been made to decipher the biology of LAT1-4F2hc. While a complete scenario has not yet been painted, a combination of bioinformatics, in vitro, and animal experiments has revealed some previously unknown aspects of LAT1-4F2hc transport mechanisms, substrate specificity, and regulation. These results provide a strong basis for pharmacological studies in which inhibitors of LAT1-4F2hc, such as JPH203. JPH203 can act well on a variety of tumor cells. Its phase I clinical trial in humans is a great milestone for researchers and patients.

However, the study of LAT1-4F2hc is still rare and not thorough in the field of urinary tumors. The results obtained so far are not fully in line with the fact that LAT1-4F2hc should play a prominent role in the field of urinary cancers. Its transport, regulation of expression/function, effects of posttranslational modifications on its stability/activity, interactions with other amino acid transporters and upstream and downstream genes, reaction with chemotherapy sensitivity/resistance, relationship with immunotherapy of sensitivity/resistance, is worthy for further research. In addition, Whether the phase I or II clinical trials of JPH203 in patients with urinary tumors can improve the prognosis of urinary tumors and whether there are corresponding biomarkers that can be used to predict the sensitivity and prognosis of inhibitors are also worthy of study. 

As a future direction, we are currently pursuing the utility of LAT1 as a biomarker in urological tumors. In recent years, the usefulness of liquid biopsy has been suggested in clinical practice. The expression of LAT1 in blood, including CTCs, ctDNA, and Exosome, is currently being examined through collaborative research.

We hope to prove its usefulness not only as an inhibitor but also as a companion di-agnostic agent in the near future. 

In conclusion, LAT1-4F2hc plays an important role in the diagnosis, treatment, and prognosis assessment of urinary system tumors. Cancer-related amino acid transporters may change the diagnostic and treatment strategy of urological tumors in near future. 

## Figures and Tables

**Figure 1 cancers-14-00229-f001:**
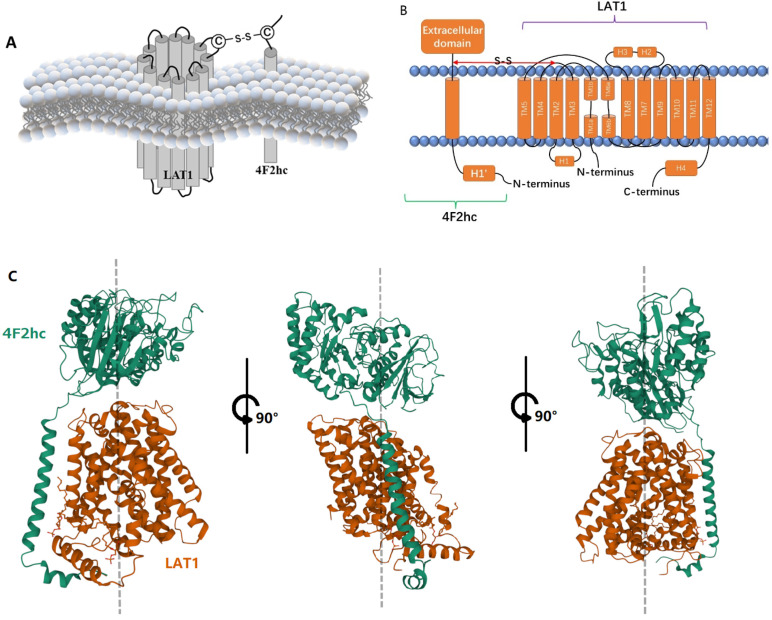
Structure of LAT1-4F2hc Complex: (**A**) Hypothetical model of the complex of LAT1 and 4F2hc; (**B**) LAT1 has 12 transmembrane units, while 4F2hc has only one. The two are covalently connected by disulfide bonds; (**C**) FIG1 (**C**) Images created using Mol*, the PDB ID: 6IRS, Structure of the human LAT1-4F2hc heteromeric amino acid transporter complex. [19], Mol* (D. Sehnal, S. Bittrich, M. Deshpande, R. Svobodová, K. Berka, V. Bazgier, S. Velankar, S.K. Burley, J. Koča, A.S. Rose (2021) Mol* Viewer: modern web app for 3D visualization and analysis of large biomolecular structures. Nucleic Acids Research. doi: 10.1093/nar/gkab314 [21]), and RCSB PDB.

**Figure 2 cancers-14-00229-f002:**
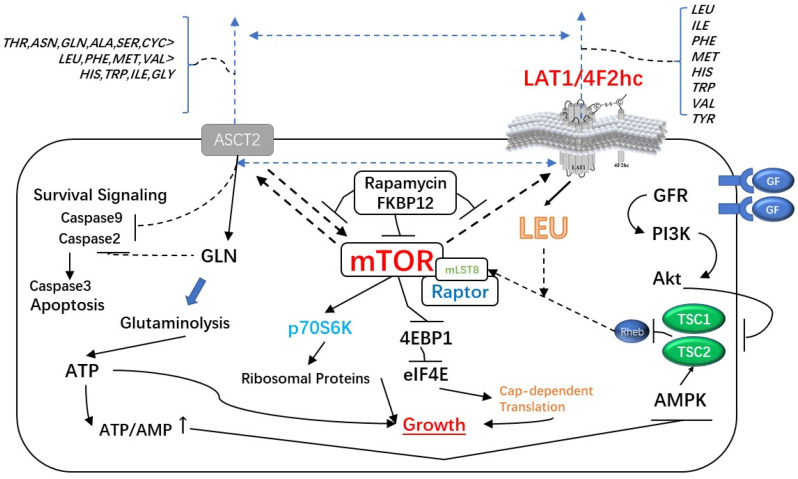
The Major Signaling Pathways Affected by LAT1-4F2hc Complex: The LAT1-4F2HC complex not only enhances mTORC1 activity but also enhances MYC and EZH2 signaling pathways. Moreover, it can affect the proliferation of cancer cells through AKT, MAPK and cell cycle-related P21 and P27 signaling pathways.

**Figure 3 cancers-14-00229-f003:**
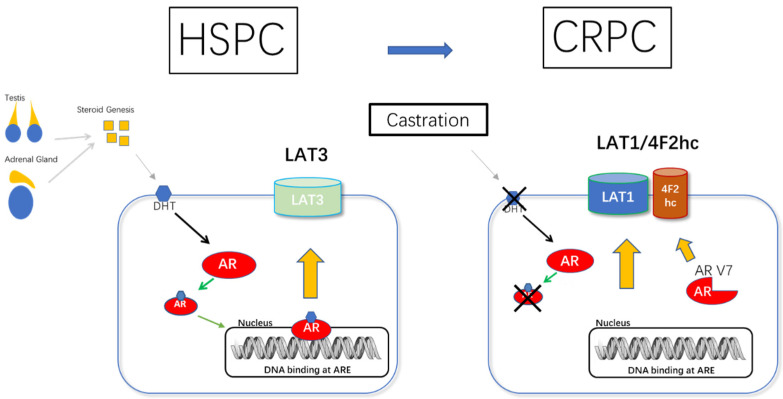
LAT Expression is Coordinately Regulated During Prostate Cancer Progression: Proposed model of LAT1-4F2hc/LAT3 in HSPC to CRPC. As HSPC progresses to CRPC, AR acts in reverse to cause low expression of LAT3 and high expression of LAT1.

**Figure 4 cancers-14-00229-f004:**
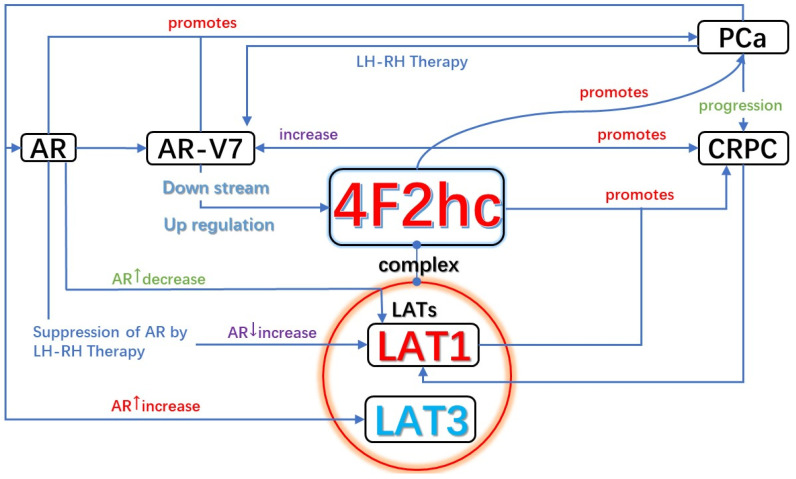
Relationship of LAT1-4F2hc and PCa & CRPC: The relationship between LATx-4F2hc and AR(AR-V7) and different stages of prostate cancer. Reduced androgen receptor signaling and variation of androgen receptors may result in decreased LAT3 expression and higher LAT1 expression.

**Table 1 cancers-14-00229-t001:** LAT1-4F2hc and Common Tumors.

	Cancer Types	Cell Lines	Downstream Effects of LAT1/4F2hc	Other Related Factors	References
LAT1-4F2hc Complex	NSCLC	A549, H1299	Mice with smaller tumors, lower leucine absorption, lower mTORC1 activity, amino acid stress, lower proliferation, and lower EZH2 expression and activity	Ki-67, VEGF, CD31, CD34, HIF-1a, mTOR, ASCT2	[27,52,58,59,60,61,62,63]
	Gastric cancer	SGC-7901, MKN-45, MGC-803, CRL-5974	Deceases in proliferation, migration and invasion	Ki-67	[25,64,65,66,67,68,69]
	Pancreatic cancer	MIA, Paca-2	Reductions in mTORC1 activity, decreases in proliferation and angiogenesis	Ki-67, VEGF, c-Myc, CD147	[50,70,71,72,73,74]
	Biliary tract cancer	KKU-M055, KKU-M213	JPH203 first in human phase I clinical trial. Well-tolerated.	Ki-67	[75,76,77,78,79,80,81]
	Ovarian cancer	SKOV3, IGROV1, A2780, OVCAR-3	Decreases in proliferation	ASCT2, SN2, p70S6K, LAT2	[82,83,84,85]
	Breast cancer & TNBC	MCF-7, ZR-75, MDA-MB-232	Decreases in proliferation	ADS, HER2, TN, Ki-67, ER, PgR	[45,86,87,88,89,90]

**Table 2 cancers-14-00229-t002:** LAT1-4F2hc and Urological Tumors.

LAT1-4F2hc and Urological Tumors										
Cancer Types	Cell Lines	Expression		Inhibitors		Be Inhibited Downstream Effects		Other Related Factors	Meanings	References
		LAT1	4F2hc	LAT1	4F2hc	LAT1	4F2hc			
Prostate cancer	LNCAP	↑(Not express [95])	↑	BCH, JPH203, R1881, ESK242	BCH, JPH203, R1881, AR-V7 knockdown	Lower leucine absorption, Lower mTORC1 activity, Amino acid stress, Down regulation of ATF4-mediated genes, Reduced tumor metastasis ability in PC3-CRPC metastatic tumor mouse model.	Lower proliferation, higher apoptosis, and several gene expression changes.	LAT3, ATF4, ASCT1, ASCT2, SKP-2, ADT (LH-RH Therapy), y+LAT2, mTORC1, Ki-67, AR, AR-V7, SLFN5	A biomarker of PCa. Associated with high Gleason score, improving drug delivery in PCa cells. Specific antibodies to LAT1 can inhibit tumor growth. Expression changes when hormone ablation and in metastatic lesions. The expression levels of LAT1 and 4F2hc suggest different prognosis respectively.	[16,24,44,57,94,95,97,111]
	LNCAP95	↑	↑							
	C4-2	↑	↑							
	PC3	↑	↑							
	DU145	↑	N/A							
	VCAP	↑	↑							
Renal cancer	Caki-1	↑	N/A	JPH203	JPH203	Lower mTORC1 activity, Reduced p70S6K and 4E-BP1.	N/A	S6 ribosomal protein (Ser-235/236)	An RCC biomarker for diagnosis and treatment. Related to the poorer differentiation, associated with local invasion and microscopic vascular invasion. LAT1-mRNA is a target for therapy. Promising prognostic markers. High LAT1 expression suggests a poor prognosis (OS & PFS).	[42,98]
	ACHN	↑	N/A							
	ccRCC tissue	↑	→(by mRNA detection)							
Bladder cancer	T24	↑	↑	BCH, JPH203, SiLAT1	BCH	Cell growth inhibition, inhibit phosphorylation of MAPK/Erk, AKT, p70S6K, and 4EBP-1. Decreases in migration and invasion activities.	Reduced Leucine intake and tumor cell growth.	P27, Ki-67, IGFBP-5	An independent prognostic factor. Associated with the tumor stage.	[93,109,110,112]
	5637	↑	N/A							

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
