# Peer review of "Contribution of LAT1-4F2hc in Urological Cancers via Toll-like Receptor and Other Vital Pathways"

_cancers, 2022, doi:10.3390/cancers14010229_

Round 1

Reviewer 1 Report

Dear Authors

Thank you very much for your manuscript submission. Your work is well-designed; however a major revision is needed as below:

    1. Please read and add the following papers to the References section of the manuscript. These papers represent invaluable data and information which can be used to have a more fruitful manuscript:

      The Druggability of Solute Carriers. J Med Chem. 2020 Apr 23;63(8):3834-3867. doi: 10.1021/acs.jmedchem.9b01237. Epub 2019 Dec 19. PMID: 31774679.

      An Overview of Cell-Based Assay Platforms for the Solute Carrier Family of Transporters. Front Pharmacol. 2021 Aug 10;12:722889. doi: 10.3389/fphar.2021.722889. PMID: 34447313; PMCID: PMC8383457.

      Structure of the human LAT1-4F2hc heteromeric amino acid transporter complex. Nature. 2019 Apr;568(7750):127-130. doi: 10.1038/s41586-019-1011-z. Epub 2019 Mar 13. PMID: 30867591.

      Heteromeric Solute Carriers: Function, Structure, Pathology and Pharmacology. Adv Exp Med Biol. 2021;21:13-127. doi: 10.1007/5584_2020_584. PMID: 33052588.

    2. I recommend to add a subtitle regarding the association between Toll-like receptors and SLC7A5 and its probable  importance relating to urological cancers. Please read and add the following papers to the References section of the manuscript:

      l-Leucine influx through Slc7a5 regulates inflammatory responses of human B cells via mammalian target of rapamycin complex 1 signaling. Mod Rheumatol. 2019 Sep;29(5):885-891. doi: 10.1080/14397595.2018.1510822. Epub 2018 Sep 5. PMID: 30092695.

      Putative roles of SLC7A5 (LAT1) transporter in pain. Neurobiol Pain. 2020 Jun 30;8:100050. doi: 10.1016/j.ynpai.2020.100050. PMID: 32715162; PMCID: PMC7369351.

      Toll-Like Receptors: General Molecular and Structural Biology. J Immunol Res. 2021 May 29;2021:9914854. doi: 10.1155/2021/9914854. PMID: 34195298; PMCID: PMC8181103.

    3. I recommend to use illustrative facilities such as PDB, etc. In this regard, please read and add the following paper to the Reference section of the manuscript. This paper represents a wide range of illustrative software tools and websites which can be employed in your manuscript.

      Writing a strong scientific paper in medicine and the biomedical sciences: a checklist and recommendations for early career researchers. Biol Futur. 2021 Dec;72(4):395-407. doi: 10.1007/s42977-021-00095-z. Epub 2021 Jul 28. PMID: 34554491.

    4. Please do add an additional column to the tables regarding the references. The present form of the tables is very confusing.

    5. In accordance with recommended papers, the Conclusion section can be improved.

Author Response

Reviewer #1:

Thank you very much for your manuscript submission. Your work is well-designed; however a major revision is needed as below: Please read and add the following papers to the References section of the manuscript. These papers represent invaluable data and information which can be used to have a more fruitful manuscript:

  1. The Druggability of Solute Carriers. J Med Chem. 2020 Apr 23;63(8):3834-3867. doi: 10.1021/acs.jmedchem.9b01237. Epub 2019 Dec 19. PMID: 31774679.
  2. An Overview of Cell-Based Assay Platforms for the Solute Carrier Family of Transporters. Front Pharmacol. 2021 Aug 10;12:722889. doi: 10.3389/fphar.2021.722889. PMID: 34447313; PMCID: PMC8383457.
  3. Structure of the human LAT1-4F2hc heteromeric amino acid transporter complex. Nature. 2019 Apr;568(7750):127-130. doi: 10.1038/s41586-019-1011-z. Epub 2019 Mar 13. PMID: 30867591.
  4. Heteromeric Solute Carriers: Function, Structure, Pathology and Pharmacology. Adv Exp Med Biol. 2021;21:13-127. doi: 10.1007/5584_2020_584. PMID: 33052588.

Response:

We thank the Reviewer #1 for careful reviewing and kindly providing very helpful comments to improve our manuscript.

We have cited above manuscripts in Ref 19(#3), 20(#4), 34(#1), 112(#2) with the related information.

On page 2 line 92-Page 3 line 98

“The contact between 4F2hc and LAT1 is limited to one side of LAT1, while TM1 and TM6 of LAT1 are construction switches, which are essential for the alternate entry transport mechanism of the LeuT-fold transporters, and their positions are far away from the coordination of 4F2hc. Therefore, 4F2hc seems to stabilize the scaffold domain of LAT1 in the membrane, which may contribute to the local conformational shift of gating elements (such as TM1, TM2, TM6, and TM10) during alternate entry cycles. [18-20] (Figure 1. A, B, and C).”

On page 4 line 134-137

“Due to its own transport characteristics, the LAT1-4F2hc complex often plays a key role in drug absorption, distribution and toxicity by mediating drug transmembrane transport, and often represents unexpected off-target of drugs[34].”

On page 5 line 169-174

“The LAT1-4F2hc complex is also closely related to tumor glutamine metabolism. The amount of glutamine required by cancer cells exceeds the supply produced by en-dogenous synthesis, resulting in the up-regulation of glutamine metabolism in many carcinogenic changes. LAT1-4F2hc complex controls the flux of glutamine and other amino acids involved in glutaminolysis and glutamine-regulated homeostasis[34]. LAT1-4F2hc complex exchanges Gln for leucine and other amino acids, which can lead to mTOR activation.”

I recommend to add a subtitle regarding the association between Toll-like receptors and SLC7A5 and its probable importance relating to urological cancers. Please read and add the following papers to the References section of the manuscript:

Response:

We thank the Reviewer #1 for practical comment. We have changed the title as follows. 

“Contribution of LAT1-4F2hc in Urological Cancers via Toll-like receptor and other vital pathways.”

And, also, we have changed the abstract emphasizing more on the connection between mTOR and AR pathway as follows.     

On page 1 line 12-14

“LAT1 plays a role in transporting essential amino acids including leucine, which regulates the mTOR signaling pathway. However, the importance of SLCs is still not well known in the field of urological cancer.”

On page 1 line 21-26

Leucine, one of the essential amino acids, has recently been found to be closely associated with cancer, which activate mTOR signaling pathway. The transport of leucine into cells requires an L-type amino acid transporter protein 1, LAT1 (SLC7A5), which requires the 4F2 cell surface antigen heavy chain (4F2hc, SLC3A2) to form a heterodimeric amino acid transporter protein complex. Recent evidence identified 4F2hc as a specific downstream target of the androgen receptor splice variant 7 (AR-V7).

  1. l-Leucine influx through Slc7a5 regulates inflammatory responses of human B cells via mammalian target of rapamycin complex 1 signaling. Mod Rheumatol. 2019 Sep;29(5):885-891. doi: 10.1080/14397595.2018.1510822. Epub 2018 Sep 5. PMID: 30092695.

  1. Putative roles of SLC7A5 (LAT1) transporter in pain. Neurobiol Pain. 2020 Jun 30;8:100050. doi: 10.1016/j.ynpai.2020.100050. PMID: 32715162; PMCID: PMC7369351.

  1. Toll-Like Receptors: General Molecular and Structural Biology. J Immunol Res. 2021 May 29;2021:9914854. doi: 10.1155/2021/9914854. PMID: 34195298; PMCID: PMC8181103.

We thank the Reviewer #1 for careful reviewing and kindly providing very helpful comments to improve our manuscript.

We have cited above manuscripts in Ref 36(#1), 35(#2), 37(#3) with the related information.

On page 4 line 138-155

“3. LAT1/4F2hc and Human Diseases (Pain & Inflammation)

Existing studies have found that LAT1-4F2hc complex is widely associated with human diseases, such as inflammation, pain, hypoxia, and tumors[35-37].

Inhibition of LAT1 eliminated mTORC1 activation, plasmablast differentiation, and CpG (toll-like receptor TLR9 ligand)-stimulated B cell production of IgG and inflam-matory cytokines. The influx of L-leucine through LAT1 regulates the activity of mTORC1 and the immune response of human B cells[36,37]. Among the most common nociceptive pathways, LAT1 may be a feasible new target for pain. LAT1 expression and regulation link it to key cell types and pathways related to pain. LAT1 regulates the Wnt/frizzled/β-catenin signal transduction pathway. The LAT1-4F2hc complex may also be involved in pain pathways related to T cells and B cells. The expression of LAT1 in-duces the activation of the mammalian target of rapamycin (mTOR) signal axis, which is related to inflammation and neuropathic pain. Similarly, hypoxia and tumors can induce the activation of hypoxia-inducible factor 2α, which not only promotes the expression of LAT1 but also promotes the activation of mTORC1[35]. As the common node of the T cell, B cell, and mTOR pathway, LAT1-4F2hc plays a vital role in human diseases. It has also received increasing attention as an important target for autoimmune diseases, chronic pain diseases, and tumors.“

I recommend to use illustrative facilities such as PDB, etc. In this regard, please read and add the following paper to the Reference section of the manuscript. This paper represents a wide range of illustrative software tools and websites which can be employed in your manuscript.

We thank the Reviewer #1 for careful reviewing and kindly providing very helpful comments to improve our manuscript.

We have added the following figure of LAT1-4F2hc created by PDB.

(C)Images created using Mol*, the PDB ID:6IRS, Structure of the human LAT1-4F2hc heteromeric amino acid transporter complex.[19], Mol* (D. Sehnal, S. Bittrich, M. Deshpande, R. Svobodová, K. Berka, V. Bazgier, S. Velankar, S.K. Burley, J. Koča, A.S. Rose (2021) Mol* Viewer: modern web app for 3D visualization and analysis of large biomolecular structures. Nucleic Acids Research. doi: 10.1093/nar/gkab314), and RCSB PDB. 

Writing a strong scientific paper in medicine and the biomedical sciences: a checklist and recommendations for early career researchers. Biol Futur. 2021 Dec;72(4):395-407. doi: 10.1007/s42977-021-00095-z. Epub 2021 Jul 28. PMID: 34554491.

We thank the Reviewer #1 for careful reviewing and kindly providing very helpful comments to improve our manuscript.

We have read the article and modified accordingly.

Please do add an additional column to the tables regarding the references.

The present form of the tables is very confusing.

Response:

We thank the Reviewer #1 for careful reviewing and kindly providing very helpful comments to improve our manuscript.

We have added the extra column for the reference in table 1 and 2.

We re-edited table 2 to make it easier to be understood.

In accordance with recommended papers, the Conclusion section can be improved.

Response:

We thank the Reviewer #1 for careful reviewing and kindly providing very helpful comments to improve our manuscript.

We have added the following text in the conclusion section.

On page 3 line 476-479.

“The LAT1-4F2hc complex mediates a variety of pathways, such as T cells, B cells, and mTOR pathways, and is also closely related to Toll-like receptors and vascular en-dothelial growth factors. This has caused the LAT1-4F2hc complex to become a common factor in many diseases, such as autoimmune diseases, pain, tumors.”

On Page 3 line 502-510

“As a future direction, we are currently pursuing the utility of LAT1 as a biomarker in urological tumors. In recent years, the usefulness of liquid biopsy has been suggested in clinical practice. The expression of LAT1 in blood, including CTCs, ctDNA, and Ex-osome, is currently being examined through collaborative research.

We hope to prove its usefulness not only as an inhibitor but also as a companion di-agnostic agent in the near future.

In conclusion, LAT1-4F2hc plays an important role in the diagnosis, treatment, and prognosis assessment of urinary system tumors. Cancer-related amino acid transporters may change the diagnostic and treatment strategy of urological tumors in near future.”

Reviewer 2 Report

Xue Zhao et al uncovered the role of LAT1-4F2hc in Urological malignancies.

Points to be addressed:

1) The rationale of why the authors came up with this review.

2) What is the information that is not exactly available that motivated the authors to come up with this information. What are the current caveats and how do the authors highlight the current research in answering them? If not they need to address in future directions.

3) The authors allude to angiogenesis and immunotherapy. I think that this point can be slightly expanded especially focusing on renal cancer: tumors grow and evolve through constant crosstalk with the surrounding microenvironment, and emerging evidence indicates that angiogenesis and immunosuppression frequently occur simultaneously in response to this crosstalk. Accordingly, strategies combining anti-angiogenic therapy and immunotherapy seem to have the potential to tip the balance of the tumor microenvironment and improve treatment response: please refer to PMID: 33203154 end expand. 

4) anti-angiogenesis and immunotherapy represent novel paradigms to envision tailored approaches in renal cell-carcinoma. Since there is a correlation of LAT1/4F2hc expression with pathological grade, proliferation and Angiogenesis the authors need to substantiate.

5) The authors need to highlight what new information the review is providing to enhance the research in progress.

Author Response

Reviewer #2:

Points to be addressed:

  • The rationale of why the authors came up with this review.

Response:

We thank the Reviewer #2 for careful reviewing and kindly providing very helpful comments to improve our manuscript.

  1. We have raised the rationale as follows and added on the text on page 2 line 78-83

However, the importance of SLCs is still not well known in the field of urological cancer. In particular, LAT1 is a target of FACBC PET[15], which has important imaging implications in prostate cancer, following PSMA PET. Recently, 4F2hc, which binds to LAT1, has been identified as a specific downstream signal of AR-V7, a cause of castra-tion resistance[16]. JPH203, a specific inhibitor of LAT1, has already completed Phase I clinical trials in Japan and may be applied to prostate cancer in the future[17].

  • What is the information that is not exactly available that motivated the authors to come up with this information. What are the current caveats and how do the authors highlight the current research in answering them? If not they need to address in future directions.

Response:

We thank the Reviewer #2 for careful reviewing and kindly providing very helpful comments to improve our manuscript.

  1. on page 6 line 198-207, we have added the information related to the connection between FACBC PET and LAT1

“Trans-1-amino-3-18F-fluorocyclobutanecarboxylic-acid (anti-[18F]-FACBC) is an amino acid PET tracer, which shows good prospects in visualizing PCa[90]. The tracer is used for the evaluation of l-amino acid transport, LAT1 is known to be the primary target of FACBC [15]. In 2016, 18F-FACBC has been approved by the US Food and Drug Administration (FDA) and the European Commission (EC) to detect PCa in patients with elevated PSA after previous treatments[14]. Approval is based on encouraging diagnostic performance and histologically confirmed data from patients with biochemical relapse[91]. Recently, it was included in the National Comprehensive Cancer National (NCCN) guidelines for the treatment of patients with recurrent PCa. The usefulness of LAT1 in PET imaging has already been validated in clinical practice.”

  1. On page 15 line 502-510, we have added the future perspective

As a future direction, we are currently pursuing the utility of LAT1 as a biomarker in urological tumors. In recent years, the usefulness of liquid biopsy has been suggested in clinical practice. The expression of LAT1 in blood, including CTCs, ctDNA, and Ex-osome, is currently being examined through collaborative research.

We hope to prove its usefulness not only as an inhibitor but also as a companion di-agnostic agent in the near future.

In conclusion, LAT1-4F2hc plays an important role in the diagnosis, treatment, and prognosis assessment of urinary system tumors. Cancer-related amino acid transporters may change the diagnostic and treatment strategy of urological tumors in near future.

3) The authors allude to angiogenesis and immunotherapy. I think that this point can be slightly expanded especially focusing on renal cancer: tumors grow and evolve through constant crosstalk with the surrounding microenvironment, and emerging evidence indicates that angiogenesis and immunosuppression frequently occur simultaneously in response to this crosstalk. Accordingly, strategies combining anti-angiogenic therapy and immunotherapy seem to have the potential to tip the balance of the tumor microenvironment and improve treatment response: please refer to PMID: 33203154 end expand.

Response:

We thank the Reviewer #2 for careful reviewing and kindly providing very helpful comments to improve our manuscript.

  1. On page 9 line 305-340, we have added following text

“Tumors grow and evolve through continuous crosstalk with the surrounding microenvironment. New evidence shows that angiogenesis and immunosuppression often occur simultaneously to deal with this crosstalk[98]. At present, one strategy to achieve a higher clinical response in the study of renal cell carcinoma is to produce a more effective anti-tumor contraction by combining multiple immune checkpoints. But the toxicity profile is higher[99]. T cells can shape tumor blood vessels and tumor endothelial cells, prevent the recruitment and infiltration of effector immune cells while remodeling ECM, and further inhibit the migration and infiltration of functional immune cells. The tumor vascular system actively participates in immunosuppression. The abnormal pathophysiological mechanism of tumor vessels can lead to the production of immunosuppressive molecules and inhibit the function of effective T cytotoxic cells. At the same time, the production of chemokines and cytokines promotes the differentiation and activation of immunosuppressive cells. These cells can also inhibit the activity of cytotoxic T cells. On the contrary, in the blood vessels, these mechanisms also down-regulate a variety of adhesion molecules, which are very important for the rolling, adhesion, and transport of T cells into the cancer environment. The normal tumor vascular system can improve T cell infiltration, enhance immune response, stop the immunosuppressive environment, make it a more immunoactivated phenotype, and work together with cancer immunotherapy. Anti-vascular endothelial growth factor receptor (anti-VEGFR) is the first to realize the normalization and functional recovery of tumor vascular system by tissue perfusion and reducing intratumoral hypoxia[98].  In the current studies of cancers [70,100,101], angiogenesis in vitro / in vivo experiments was inhibited by eliminating the function or expression of LAT1. It regulates proliferation, translation, and angiogenesis VEGF-A signal[101]. “

  1. On page 17 line 443-446, we have added following text

“However, the LAT1-4F2hc complex is closely related to angiogenesis[40,70,80,100,101,104-107]. This makes it possible for the LAT1-4F2hc complex to improve the effectiveness of cancer immunotherapy by improving immune vascular crosstalk[98].”

4) anti-angiogenesis and immunotherapy represent novel paradigms to envision tailored approaches in renal cell-carcinoma. Since there is a correlation of LAT1/4F2hc expression with pathological grade, proliferation and Angiogenesis the authors need to substantiate.

On page 9 line 327 - 340

“LAT1 is a central transporter of essential amino acids in human umbilical vein endo-thelial cells[102]. LAT1 also mediated miR-126 on primary human lung microvascular endothelial cells’ angiogenesis via regulation of mTOR signaling[103]. LAT1 expression correlated significantly with CD98, VEGF, CD34 expression, and microvessel density in the primary and metastatic sites of tumors[40,80,104-107]. VEGF and CD34 are also re-lated to angiogenesis. These studies further revealed the dual role of LAT1-4F2hc in tumor cells and stromal endothelial cells. The therapeutic inhibition of LAT1-4F2hc may provide an ideal choice for strengthening anti-angiogenesis therapy. Lat1-4f2hc is a potential therapeutic target for anti-tumor angiogenesis and maintenance of the normal vascular system. Therefore, the combination of antiangiogenic therapy and immuno-therapy seems to have the potential to break the balance of the tumor microenvironment and improve the treatment response of renal cell carcinoma. It can be a novel paradigm to envision tailored approaches in renal cell-carcinoma and other urological tumors.”

5) The authors need to highlight what new information the review is providing to enhance the research in progress.

We thank the Reviewer #2 for careful reviewing and kindly providing very helpful comments to improve our manuscript.

We have changed the abstract emphasizing more on the connection between mTOR and AR pathway as follows.     

On page 1 line 12-14

“LAT1 plays a role in transporting essential amino acids including leucine, which regulates the mTOR signaling pathway. However, the importance of SLCs is still not well known in the field of urological cancer.”

On page 1 line 21-26

Leucine, one of the essential amino acids, has recently been found to be closely associated with cancer, which activate mTOR signaling pathway. The transport of leucine into cells requires an L-type amino acid transporter protein 1, LAT1 (SLC7A5), which requires the 4F2 cell surface antigen heavy chain (4F2hc, SLC3A2) to form a heterodimeric amino acid transporter protein complex. Recent evidence identified 4F2hc as a specific downstream target of the androgen receptor splice variant 7 (AR-V7).

In addition to the references suggested by experts, we have also added the following references.

LIST:

  1. (18)F-Facbc in Prostate Cancer: A Systematic Review and Meta-Analysis. Cancers (Basel) 2019, 11, doi:10.3390/cancers11091348.

  1. Amino acid transporter expression and 18F-FACBC uptake at PET in primary prostate cancer. Am J Nucl Med Mol Imaging 2021, 11, 250-259.

  1. Biodistribution and radiation dosimetry of the synthetic nonmetabolized amino acid analogue anti-18F-FACBC in humans. J Nucl Med 2007, 48, 1017-1020, doi:10.2967/jnumed.107.040097.

  1. Multisite Experience of the Safety, Detection Rate and Diagnostic Performance of Fluciclovine ((18)F) Positron Emission Tomography/Computerized Tomography Imaging in the Staging of Biochemically Recurrent Prostate Cancer. J Urol 2017, 197, 676-683, doi:10.1016/j.juro.2016.09.117.

  1. Nivolumab plus Ipilimumab versus Sunitinib in Advanced Renal-Cell Carcinoma. N Engl J Med 2018, 378, 1277-1290, doi:10.1056/NEJMoa1712126.

  1. Amino acid transporter LAT1 in tumor-associated vascular endothelium promotes angiogenesis by regulating cell proliferation and VEGF-A-dependent mTORC1 activation. J Exp Clin Cancer Res 2020, 39, 266, doi:10.1186/s13046-020-01762-0.

  1. LAT1 is a central transporter of essential amino acids in human umbilical vein endothelial cells. J Pharmacol Sci 2014, 124, 511-513, doi:10.1254/jphs.13255sc.

  1. MicroRNA-126-3p Inhibits Angiogenic Function of Human Lung Microvascular Endothelial Cells via LAT1 (L-Type Amino Acid Transporter 1)-Mediated mTOR (Mammalian Target of Rapamycin) Signaling. Arterioscler Thromb Vasc Biol 2020, 40, 1195-1206, doi:10.1161/atvbaha.119.313800.

  1. Prognostic significance of L-type amino acid transporter 1 (LAT1) and 4F2 heavy chain (CD98) expression in early stage squamous cell carcinoma of the lung. Cancer Sci 2009, 100, 248-254, doi:10.1111/j.1349-7006.2008.01029.x.

  1. Correlation of angiogenesis with 18F-FMT and 18F-FDG uptake in non-small cell lung cancer. Cancer Sci 2009, 100, 753-758, doi:10.1111/j.1349-7006.2008.01077.x.

106.Correlation of L-methyl-11C-methionine (MET) uptake with L-type amino acid transporter 1 in human gliomas. J Neurooncol 2010, 99, 217-225, doi:10.1007/s11060-010-0117-9.

107.Relation of LAT1/4F2hc expression with pathological grade, proliferation and angiogenesis in human gliomas. BMC Clin Pathol 2012, 12, 4, doi:10.1186/1472-6890-12-4.

  1. Reevaluating the Substrate Specificity of the L-Type Amino Acid Transporter (LAT1). J Med Chem 2018, 61, 7358-7373, doi:10.1021/acs.jmedchem.8b01007

3) The authors allude to angiogenesis and immunotherapy. I think that this point can be slightly expanded especially focusing on renal cancer: tumors grow and evolve through constant crosstalk with the surrounding microenvironment, and emerging evidence indicates that angiogenesis and immunosuppression frequently occur simultaneously in response to this crosstalk. Accordingly, strategies combining anti-angiogenic therapy and immunotherapy seem to have the potential to tip the balance of the tumor microenvironment and improve treatment response: please refer to PMID: 33203154 end expand. 

4) anti-angiogenesis and immunotherapy represent novel paradigms to envision tailored approaches in renal cell-carcinoma. Since there is a correlation of LAT1/4F2hc expression with pathological grade, proliferation and Angiogenesis the authors need to substantiate.

Round 2

Reviewer 1 Report

Dear Authors

Thank you very much for your rigorous revision. Accept

Reviewer 2 Report

The authors have clarified several of the questions I raised in my previous review. Most of the major problems have been addressed by this revision.